# New Insights on the Role of Marinobufagenin from Bench to Bedside in Cardiovascular and Kidney Diseases

**DOI:** 10.3390/ijms241311186

**Published:** 2023-07-06

**Authors:** Nazareno Carullo, Giuseppe Fabiano, Mario D’Agostino, Maria Teresa Zicarelli, Michela Musolino, Pierangela Presta, Ashour Michael, Michele Andreucci, Davide Bolignano, Giuseppe Coppolino

**Affiliations:** Renal Unit, “Magna Graecia” University of Catanzaro, 88100 Catanzaro, Italy

**Keywords:** Marinobufagenin (MBG), chronic kidney disease, end-stage renal disease (ESRD), nervous and immune systems

## Abstract

Marinobufagenin (MBG) is a member of the bufadienolide family of compounds, which are natural cardiac glycosides found in a variety of animal species, including man, which have different physiological and biochemical functions but have a common action on the inhibition of the adenosine triphosphatase sodium-potassium pump (Na+/K+-ATPase). MBG acts as an endogenous cardiotonic steroid, and in the last decade, its role as a pathogenic factor in various human diseases has emerged. In this paper, we have collated major evidence regarding the biological characteristics and functions of MBG and its implications in human pathology. This review focused on MBG involvement in chronic kidney disease, including end-stage renal disease, cardiovascular diseases, sex and gender medicine, and its actions on the nervous and immune systems. The role of MBG in pathogenesis and the development of a wide range of pathological conditions indicate that this endogenous peptide could be used in the future as a diagnostic biomarker and/or therapeutic target, opening important avenues of scientific research.

## 1. Introduction

This review focuses on the recent discoveries on the role of Marinobufagenin (MBG) in CV and kidney diseases. MBG is one of the more interesting molecules belonging to the family of bufadienolides. They are part of the cardiac glycoside [1,2,3] group of molecules that have significant physiological and biochemical differences but share the ability to inhibit the adenosine triphosphatase sodium–potassium pump (Na^+^/K^+^-ATPase), an enzyme which is ubiquitous in cell membranes. They play a role in positive cardiac inotropism, as they have natriuretic and vasoconstrictive properties. Bufadienolides were initially found in several animal species, characterized by different phylogeny, suggesting their relevance in evolution (Table 1). We highlight, in particular, the role of MBG as a cardiotonic steroid in humans, as they have relevant roles in many different clinical conditions characterized by body fluid volume expansion, such as pre-eclampsia (PE), hypertension, heart failure and chronic kidney disease (CKD) (Table 2). In recent years, a growing interest has emerged in its role as a novel potential biomarker for cardiovascular (CV) disease.

## 2. Biochemical Structure and Production

The cardiac glycosides include bufadienolides and cardenolides, which substantially differ from each other in both biochemical structure and the cellular [1,2] mechanism of action. Bufadienolides are steroid compounds with a δ-lactone ring at carbon C_17_, primarily synthesized from cholesterol as a precursor via the mevalonate-independent pathway [28]. They have a double unsaturated six-membered lactone ring while cardenolides have an unsaturated five-membered lactone ring [29]. However, at present, the bufadienolide biosynthesis mechanism is still unknown [26].

Bufadienolides are found in both animals and plants [30,31] and those of mammalian origin are produced by the placenta [32] and adrenal cortex [33] under the control of the bile acid CYP27A1 enzyme, although other production sites cannot be excluded [34]. Bufadienolides are eliminated unchanged via renal excretion [13]. The bufadienolides are so named because they are extracted from the venom of a common toad, *bufo marinus* or *rhinella marina*, both members of the Bufonidae family [35]. Skin [36] and parotoid gland secretions [37] of *bufo marinus* are considered the main natural sources of MBG. This toad species is native to South and Central America and has remained almost unchanged since the late Miocene period. Later, the species was introduced to Australia and the Oceanian islands and is currently one of the worst invasive species in many countries. Both sexes of the *Bufo* species possess huge parotid glands, stretching through the retro-orbital level and releasing a venom composed of different types of molecules, such as alkaloids, peptides, biogenicamines, steroids (bufogenins and bufotoxins) and proteins [38,39,40], which have antimicrobial activity and a defense action against potential predators [41,42,43]. The chemical and pharmacological characteristics of the secretions from the parotid gland and skin of the family Bufonidae have been studied for some time [44], and as early as 1972, 50 compounds were recognized in 39 species collected from different locations around the world [45].

### Mechanism of Action

The most studied of the bufadienolides is MBG, an endogenous mammalian natriuretic and cardiotonic compound with vasoconstrictive effects [1,46,47], which has a great affinity for the α1 isoform of Na^+^/K^+^-ATPase [48], the main form of the enzyme present in renal tubules [1]. In contrast, cardenolides act primarily on the α2 and α3 isoforms [48]. The Na^+^/K^+^-ATPase consists of an alpha subunit with catalytic action together with binding sites for ATP, cardiotonic steroids (CTS) and other ligands, as well as a beta subunit. Four α isoforms and three β isoforms are known for this enzyme. The α1β1 complex is largely present in various tissues, and the α2 isoform is mainly present in cardiac, smooth muscle and cerebral tissues. One of the peculiarities of the bufadienolides is represented by the fact that they exert a different action according to the receptor on which they act [49]. Currently, the Na^+^/K^+^-ATPase is recognized as having three major functions: as a pump, as an enzyme and as a receptor to cardiotonic steroids [50]. A “signalling” function has also been recognized, whereby the plasmalemmal Na^+^/K^+^-ATPases reside in the caveolae of cells with other key signaling proteins [51,52]. Indeed, two distinct pathways of MBG action have been described, by which MBG acts on the Na^+^/K^+^-ATPase [50]. According to the first (defined as the ionic pathway), MBG causes an altered transmembrane ion transport by inhibiting the Na^+^/K^+^ ATPase, and this, in the kidney, results in natriuresis as a physiological response to sodium load [4,46,53]. The inhibition of Na^+^/K^+^-ATPase in the vascular smooth muscle cells induces vasoconstriction [54,55] through an increase in intracellular sodium concentration and the concomitant reversal of the function of the vascular Na^+^/Ca^++^-exchanger. This results in an increased calcium influx within smooth muscle cells, consequently causing the further release of calcium from the sarcoplasmic reticulum. The result is vasoconstriction secondary to the actin–myosin interaction [50,56]. The second mode of action (the signaling function) can cause the activation of several intracellular signals, such as mitogen-activated protein kinases (MAPK) and reactive oxygen species (ROS) inducing fibrosis [50,57]. MBG has been suggested to cause cardiac [57,58] and vascular [57,59] fibrosis, simply through the activation of the above intracellular signaling cascades.

The inhibition of Na^+^/K^+^ ATPase, caused by MBG, has different effects, depending on the tissue in which it occurs. For example, in renal tubules, it stimulates natriuresis and, at the level of the proximal tubule, it promotes the internalization of the sodium pump with a reduction in the expression of the transport protein Na^+^/H^+^ (NHE3) in the apical membrane of the renal proximal tubule [60]. MBG, after binding to the enzyme Na^+^/K^+^-ATPase, slowly dissociates to induce the endocytosis of this enzyme. This reduces sodium absorption and increases sodium excretion in the proximal renal tubule. By decreasing the amount of Na^+^/K^+^-ATPase available, it also decreases the ability to respond to changes in Na^+^ and water, leading to the promotion of water retention and volume expansion [61]. In mammals, sodium stimulates the synthesis and secretion of MBG via the angiotensin/sympathetic pathway [34]. Indeed, increased sodium intake promotes angiotensin II, aldosterone and sympathetic nervous system synthesis, resulting in the stimulation of adrenal MBG synthesis and secretion [62].

In healthy young adults, 24 h urinary MBG values were strongly linked with habitual salt intake. This is confirmed by data derived from studies in rats, whose stimulation of MBG through the intake of Na^+^ or the infusion of MBG leads to cardiac hypertrophy and vascular fibrosis [24]. In one study, a four-week administration of MBG in rats caused a significant increase in plasma aldosterone and increased systolic blood pressure values. In another study on rats, MBG infusion caused renal fibrosis, subsequently attenuated by passive immunization and improving renal function [59,63]. The use of mineralocorticoid receptor antagonists (MRA) has also been shown to have a preventive effect on MBG-induced fibrosis by occupying the binding sites of the endogenous cardiotonic steroids [64].

With regard to profibrotic pathways, Drummond et al. [65] reported possible opposite relationships between MBG and the antifibrotic microRNA miR-29b-3p, which emerged from the regulation of cardiac fibrosis in a CKD murine model, because of the Na^+^/K^+^-ATPase signaling involvement in miR-29b-3p regulation. Similar results were observed in another study conducted in cardiac fibroblasts [66,67], in which cardiac tissue was obtained from rats treated with MBG or partial nephrectomy surgery, showing opposite trends between MBG and miR-29b-3p expression in relation to collagen expression and thereby the extent of fibrosis. All together, these data indicated that CTS mediate the Na^+^/K^+^-ATPase signaling-induced regulation of miR-29b-3p expression. The mechanistic phenomena underlying this crosstalk is still yet to be totally defined and are probably due to the convergent action on molecules and kinases common to several pathways.

## 3. Extraction Techniques

Several methods for assaying these compounds have been studied, mainly for the purpose of monitoring certain drugs used in traditional Chinese medicine. The main issue concerns the difficulty of obtaining access to standard material for the setup and validation of analytical methods required for MBG measurement [68,69,70,71,72]. Generally, bufadienolides are extracted through solvent treatment from the dry or fresh skin and secretions of toads. More often, chlorinated and non-chlorinated organic solvents or alcohols are used and various techniques have been developed to separate MBG from other solutes, such as classic column chromatography, thin-layer chromatography, preparative-scale high performance liquid chromatography (HPLC) and flash column chromatography [68,73,74,75,76,77,78].

One of the first extraction techniques to obtain MBG was described by Shimada et al. in 1979. He retrieved this extract by soaking 40 *Bufo marinus* toad skins in ethanol and then dividing them into an ethyl–water acetate system. Later, the aqueous component was broken down via chromatography on Amberlite XAD-4 to split the conjugated steroids. The resulting fractions were purified via HPLC or gel chromatography on Sephadex LH-20 [79]. A different technique used by Bagrov et al. was able to separate MBG from the crystallized poison of Bufo marinus using thin layer chromatography. Today, the most widely used method for chemical characterization is mass spectrometry, allowing us to differentiate these polar molecules based on their mass spectral fragmentation paths [28]. An ELISA enzyme test kit is available for the correct measurement of MBG. Enzymatic reactions can be then quantified with an automatic photometer for microplates [20].

MBG has been extracted from human plasma and urine [8,25,80,81,82,83], and there is the possibility of measuring MBG in 24h urine samples in the presence of other steroid hormones, through solid-phase dissociation-enhanced lanthanide fluorescent immunoassay, based on a 4G4 anti-MBG mouse monoclonal antibody [80].

## 4. Marinobufagenin and Chronic Kidney Disease

CKD is one of the most frequent medical conditions worldwide and, in the general population, the CKD prevalence of all five KDIGO stages is 13.4% [84]. Low estimated glomerular filtration rate (eGFR) is a strong, independent predictor of all-cause mortality and CV diseases [85], which are the first cause of morbidity and mortality in nephropathic patients. Their increased CV risk is related to both traditional (diabetes mellitus, hypertension, etc.) and non-traditional uremia-specific risk factors [86,87,88]. Given the high social, economic and health impact of CKD worldwide, new possible underlying pathogenetic mechanisms and potential markers that may allow the early identification of the development of this complex and multifaceted disease are always being investigated.

It has been shown that, in both animal models (rats and dogs) [5,89] and in humans [74] with volume-expanding conditions, there are high plasma concentration levels of MBG. Indeed, it has been widely demonstrated that MBG production increases in all conditions of sodium and fluid retention, such as essential hypertension, heart failure, pre-eclampsia (PE), salt-sensitive hypertension in Dahl salt-sensitive (DS) rats on a high NaCl intake [46,90] and CKD.

The influence of MBG in CKD and its complications were initially evaluated in animal models. Originally, the involvement of MBG in cardiac hypertrophy was investigated in a remnant model of CKD [6,7]. In a series of investigations of these remnant kidney models, it has been shown that the development of renal dysfunction is associated with an increase in circulating concentrations of MBG [91]. On the other hand, other studies have shown an increased collagen production through fibroblasts and subsequent fibrosis in experimental uremic cardiomyopathy (UC) [9]. UC is characterized by an association with left ventricular hypertrophy (LVH) and myocardial fibrosis [92]. The etiopathogenesis of UC is extremely complex and involves several factors, such as hemodynamic overload, hypertension, anemia, mineral and bone disorders, endothelial dysfunction, insulin resistance and cardiotonic steroids, as well as several circulating uremic toxins [92,93]. It was assumed that the increase in MBG concentration was secondary to renal failure-dependent volume expansion [58]. Although extracellular volume expansion is thought to be crucial for the development of UC, there is still much debate about the exact pathogenesis [94,95,96]. Immunization against MBG in partial nephrectomy animals was associated with a substantial attenuation of cardiac hypertrophy, cardiac fibrosis and the oxidant stress state [97].

The fibrotic action of MBG on the kidney was also evaluated through the infusion of MBG in rats, which led to a peritubular and periglomerular accumulation of type I collagen at the renal cortical level [61,98]. This could be triggered by the activation of Transforming Growth Factor Beta type 1 (TGF-β1) via the renin–angiotensin–aldosterone system. In MBG-treated kidneys, the profibrotic transcription factor snail (critical regulator of the epithelial–mesenchymal transition) was expressed in both medullary and cortical tubular epithelial cells. This evidence led to a new hypothesis: MBG can have a key causative role in the epithelial–mesenchymal transition [61]. The administration of MRA occupying endogenous CTS-binding sites prevents pro-fibrotic MBG effects [64]. Since CKD is a complex set of different medical conditions, Na^+^/K^+^-ATPase alterations may not always be involved in the etiopathogenesis of different causes of CKD and MRA therapy may not be effective in all nephropathic patients. Therefore, increased MBG plasma levels could be useful in identifying patients at risk of developing renal fibrosis (and beyond) and CKD progression that could benefit from MRA therapy [19].

Recently, in a single-center study of adults routinely referred for the screening of endocrine hypertension, it was shown that plasma MBG concentrations were significantly associated with albuminuria (a marker of kidney damage) and decline in renal function regardless of pre-existing CKD. These results might indicate that in this cohort of hypertensive patients, MBG could play a role as a potential marker of renal failure at follow-up and that elevated plasma levels of MBG may already precede renal failure rather than being a simple consequence of it [19].

In patients undergoing chronic hemodialysis, the altered MBG plasma values could be due to a compensatory response to the treatment itself [20] and could predict a worsening survival outcome [21]. In this patient population, left ventricular hypertrophy is extremely prevalent and, as mentioned above, contrasts the uremic cardiomyopathy. In experimental models, MBG induces marked hypertrophic changes of adult cardiac myocytes in vitro [10,99] and promotes vascular fibrosis and cardiac hypertrophy in experimental models of salt-sensitive hypertension [91]. For these reasons, we conducted a pilot observational study to investigate a possible relationship between MBG plasma levels, left ventricular (LV) geometry and cardiac dysfunction in end-stage renal disease (ESRD) patients on dialysis [22]. In this cohort of patients with ESRD, we observed that high levels of MBG reflect the structural and functional alterations of the LV. Indeed, MBG plasma levels were higher in the presence of diastolic dysfunction and this molecule demonstrated a strong diagnostic ability to discern patients with normal LV geometry, LV hypertrophy and, above all, eccentric LVH. Circulating MBG plasma levels were significantly higher in ESRD patients than those in healthy controls and were more increased in patients on peritoneal dialysis compared with those undergoing extracorporeal dialysis treatment [22]. This suggests that, in the future, MBG could play the role of biomarker for cardiac evaluation in high-risk populations. Our findings further strengthen the hypothesis that endogenous cardiotonic steroids could substantially contribute to the onset and progression of uremic cardiomyopathy. Of note, there was no correlation between MBG plasma levels and parameters related to volume status in our study [22]. In a recent study, kidney transplant recipients displayed altered MBG levels, which were influenced by sodium balance, renal impairment and the severity of LVH. Thus, MBG might also represent an important missing link between reduced graft function and pathological cardiac remodeling and may hold important prognostic value for improving cardio-renal risk assessment [100].

Moreover, we have demonstrated that higher MBG plasma levels are associated with a lower risk of intradialytic hypotensive events in patients undergoing hemodialysis [20,101,102], who are particularly considered at risk of hypotension. Indeed, about 30% of hemodialysis sessions are characterized by severe symptomatic intra-dialysis hypotension, influencing the morbidity and mortality of patients in chronic treatment. These hypotensive episodes are often due either to an altered ability to mobilize fluids from the interstitial space to the intravascular space during the hemodialysis session or to a removal of a large fluid volume in a short time. It has been noted that, in the cohort of patients on dialysis treatment, patients with lower baseline MBG values showed an approximately five-fold higher risk of severe symptomatic, intradialytic hypotension. There was an initial increase in circulating MBG levels followed by a progressive decrease until the end of treatment. This shows that patients with lower MBG plasma values reflect a lower vascular and hemodynamic tolerance, with a higher number of episodes of severe hypotension, while no significant correlation was found between MBG plasma levels and body weight reduction during dialysis treatment. This could lead to a consideration of MBG as a mediator of a compensatory mechanism, which results in an altered hemodynamic response to plasma volume reduction. Today, it seems that MBG may have an important role in identifying patients at high risk of severe intradialytic hypotensive episodes; this predictive ability has also been found in survival analyses, demonstrating that patients with lower plasma levels of MBG had a higher risk (from four to six times) of severe intra-dialysis episodes during follow-up. MBG plasma levels were the strongest time-dependent predictors of severe intradialytic hypotensive episodes between different variables [20]. The high mobilization of MBG could initially represent a protective response against the hemodynamic changes induced by the extracorporeal treatment but, in the long term, could, in any case, determine the known deleterious effects in the myocardium [20] (Figure 1). However, larger studies with more targeted-oriented surveys are necessary to confirm these data.

## 5. Marinobufagenin and CV Diseases

Most of the studies designed to evaluate the pathogenetic mechanisms by which MBG contributes to CV disease risk have been performed in animal models. Increased concentrations of circulating MBG (as a result of sodium loading or via infusion pump) caused several effects: vascular [24,59] and microcirculation [103] alterations, pressor changes [4,46,104,105] and cardiac and renal [8,9,58,61,91] fibrosis. Investigations in humans have shown elevated MBG plasma levels in many pathological conditions: heart failure [8], acute myocardial infarction (elevated urinary MBG levels) [106], primary aldosteronism [107], renal ischemia [108] and CKD [100,109].

There are several scientific pieces of evidence supporting a possible pathogenic role of bufadienolides in hypertensive conditions associated with hydro-saline retention. Firstly, increased plasma and urinary concentration levels of MBG were observed in volume expansion conditions and in hypertensive patients mediated by volume expansion, due to salt accumulation [46,104,105,110]. Secondly, the administration of bufadienolides in experimental animals causes hypertension [11,13,111]. Thirdly, in rats, the hypertension caused by the administration of deoxycorticosterone acetate and salt is reversed through the intraperitoneal injection of an MBG antagonist, resibufogenin (RBG) [12,13,112], which differs from MBG only in the absence of a hydroxyl group in the β 5 position. Finally, the use of anti-MBG antibodies in salt-loaded pregnant rats and in salt-sensitive hypertensive rats, results in the reduction in blood pressure values [113].

In recent years, several evaluations of whether MBG could be used as an early marker of CV risk have been performed. The first study designed to evaluate the possible association between blood pressure values and MBG plasma levels demonstrated an inverse relationship between diastolic blood pressure values and the urinary excretion (24 h) levels of MBG in the case of high sodium intake (16.32 g of salt per day). The natriuretic effects of MBG could represent a homeostatic mechanism to restore blood pressure values to normal, constituting a protective mechanism in healthy subjects [82]. The dietary intervention had a total duration of 12 days; therefore, that effect could represent a homeostatic mechanism in the short term. In contrast, another group has shown that MBG plasma levels are positively associated with systolic blood pressure values during the period of high sodium intake (5 weeks). In that case, it might reflect a long-term homeostatic response in which the vasoconstrictor activity of MBG could superimpose the natriuretic effect [114]. Fedorova et al. showed completely different responses from the above studies. In a cohort of men and women, there was an increase in systolic blood pressure values following dietary sodium loading without changes in both plasma and urinary MBG concentrations [81]. Therefore, the results on the relationship between blood pressure values and circulating and urinary MBG levels in humans are conflicting and require further evaluation.

Recent findings have revealed that in a state of inactivity, sodium can settle in the interstitium between the skin and organs [115]. The alteration of these deposits could also affect blood pressure values. We can say that high sodium intake correlates both with a higher production of MBG and with rigid large arteries, even in healthy subjects. These pieces of evidence were confirmed in laboratory models, as risk factors for dementia and CV events [17]. In addition to MBG, several factors act on peripheral vascular resistance, such as neurohormonal, baroreflexes and myogenic factors [49]. It is also known that sodium regulates the rigidity of endothelial cells and modifies the release of nitric oxide by altering the tone of blood vessels and blood pressure [116]. In the current state of the art, there is still not enough evidence to consider urinary MBG excretion as a predictive value of increased cardiovascular risk before the onset of the disease [24], although numerous studies have confirmed the correlation between plasma MBG levels, sodium and occurrence of arterial hypertension. Furthermore, these results proved to be more applicable to men than to women.

In other parts of the body, such as the arterial vascular smooth muscle cells, MBG generates an increase in cytosolic Ca2+ amount that causes vasoconstriction through the activation of an “ionic pathway” [4]. In addition to this “ionic pathway”, MBG can activate different intracellular signaling pathways that trigger cellular effects, such as cell proliferation, ROS genesis or the stimulation of apoptosis, also via the activation of pathways with other molecules, such as Phospholipase C-*γ* isozyme (PLC-γ), Phosphatidylinositol 3-kinase (PI-3K), IP3 receptor type 3 (IP3R), and ankyrin [117].

Specifically, it has been reported that cardiotonic steroids activate a signal cascade, which is mediated through Src, Ras, ROS, and ERKs, and promote endocytosis of the plasmalemmal Na^+^/K^+^-ATPase [10,99,118,119,120]. The activation of this cascade requires that the Na^+^/K^+^-ATPase to be in caveolae in order to proceed [121,122]. This series of signals is known to cause changes in gene expression, which can be inhibited by antioxidant molecules [7,10,120].

Arterial stiffness is well known to be related to increased CV risk and death in individuals of all ages [123], regardless of blood pressure values. Sodium intake also correlates with arterial stiffness regardless of hypertensive status [124,125], even in healthy subjects. Thus, a possible association between MBG and arterial stiffening was hypothesized. Jablonski et al. have demonstrated, in individuals with high or hypertensive blood pressures, that a positive association between MBG and carotid to femoral pulse wave velocity (cfPWV) [114] is actually the gold standard measurement of large artery stiffness [126]. The same positive correlation has also been demonstrated in young healthy women, regardless of salt intake [80]. To date, the pathogenetic mechanisms through which MBG is able to determine arterial stiffness are unknown, although MBG has already been shown to promote the development of vascular fibrosis in the aorta of rats [59]. The mechanism by which MBG promotes collagen production and deposition was studied in cultured rat smooth muscle cells and was always dependent on the inhibition of Na^+^K^+^-ATPase [58,59]. Collagen-1 production is secondary to the marked downregulation of transcription factor Friend leukemia integration-1 (Fli-1) [58,59]. In fact, it has been shown that MBG could also sub-regulate a negative regulator of collagen-1 synthesis, Fli-1. The phosphorylation of Fli-1 through the active form of Protein Kinase C Delta (PKC-d) induces the activation of a collagen gene promoter. In vitro, MBG was found to be an activator of the Fli-1 pathway in cultured fibroblasts and smooth rat muscle cells.

Another important predictor of both increased CV risk and mortality is the left ventricular mass (LVM) measured using the echocardiogram [127]. In the CARDIA study (Coronary Artery Risk Development in Young Adults), a possible positive association between LVM and sodium in 24h urine levels was shown in young adults, although this relationship was confounded by the presence of obesity [128]. From this observation, it was hypothesized that the increase in MBG plasma levels, induced by sodium intake, may be associated with an increase in LVM. Strauss et al. have observed a significant association between LVM and MBG in young adults: in this case, the relationship was independent on obesity and also on blood pressure values, suggesting a possible pathway through which MBG induces myocardial hypertrophy [25].

To further confirm the correlation between MBG plasma levels and fibrosis, it has been shown that rats with (experimental) renal impairment have increased MBG plasma values, along with cardiac and renal fibrosis. In these mouse models, MBG promotes procollagen-1 expression by cultured cardiac fibroblasts. As procollagen expression increases, collagen and procollagen-1 mRNA increase too. Considering this, MBG probably induces a direct increase in collagen expression by fibroblasts [9]. It has also been observed that in spontaneously hypertensive (SHR) and normotensive Wistar–Kyoto (WKY) mouse models, following a diet rich in sodium, hypertension and left and renal ventricular hypertrophy, due to fibrosis with the overexpression of TGF-β_1_ mRNA, were recorded. This has resulted, in both glomerular and peritubular sites, in an increase in collagen type 1 [129]. Interestingly, in animal models characterized by prolonged salt intake, diet-related profibrotic effects were eliminated through treatment with anti-MBG antibodies, without generating hemodynamic effects [130]. The administration of specific antibodies against MBG reduced aortic fibrosis and favored relaxation at the level of aortic fibers. The differences that can be attributed to the vascular component without hemodynamic changes, indicate that the possible vascular stiffening is independent of blood pressure and that the profibrotic factor generated by MBG is responsible for it [131].

Even in laboratory models treated with antibodies to MBG and previously subjected to a diet high in sodium, there was a reduction in systolic blood pressure and also a reduction in the weight of both the heart and kidneys. In these mouse models, TGF-β expression, which had previously been increased, was also downregulated after treatment with anti-MBG antibodies. The immunoneutralization of MBG resulted in the downregulation of genes involved in profibrotic expression. In addition, the normalization of renal function through creatinine clearance was also observed, probably due to the reduction in renal fibrosis, as it was accompanied by a significant decrease in kidney weight for the reduction in type I-III-IV collagen amounts. In mouse models, it has also been observed that treatment with anti-MBG antibodies reduces the development of heart failure. This was established by estimating ventricular weight via ultrasound in hypertensive rats immunoneutralized for MBG, compared with non-immunoneutralized hypertensive rats. 

Early vascular damage was also reduced after immunoneutralization. In these models, a reduction in the mRNA expression of TGF-β1, FN1, MAPK1, Col1a2, Col3a1 and Col4a1 was also noted compared with those not treated with antibodies of MBG. This demonstrated how MBG initiates TGFβ-1-signaling in cultured ventricular myocytes, through Na^+^/K^+^-ATPase signal transduction and other factors, such as tissue angiotensin II [132]. In humans, sodium restriction also reduces urinary MBG production and excretion, resulting in reduced blood pressure and aortic stiffness. Another factor may impact fibrosis, sodium concentration and MBG plasma levels: it has been observed that the sensitivity of blood pressure dietary salt intake increases with increasing age [23,133,134].

## 6. MBG in Relation to Sex and Gender Medicine with a Special Focus on Pre-Eclampsia

In the last few years, the concept of gender medicine emerged in relation to the presence of significant and underestimated effects of gender-based differences on clinical evolution and therapeutic outcomes of many diseases [135,136,137]. Despite CTS, such as MBG, playing a role in human physiopathology, regardless of sex- and gender-related aspects, there is evidence regarding specific actions occurring in sex-specific diseases, such as pregnancy diseases [68], which can be considered a valid example of gender medicine. MBG was identified as a biomarker of angiogenic imbalance in the pathogenesis of PE [47], a relatively common and potentially devastating complication of pregnancy.

PE is a progressive multisystem syndrome that is characterized by the onset, after the 20th week of gestation, of hypertension and proteinuria or by the onset of hypertension with severe organ dysfunction, with or without proteinuria [138,139,140]. A systematic review stated that 4.6% of pregnancies worldwide were complicated by PE [141], which represents the second leading cause of fetal and maternal morbidity and mortality [142,143,144,145]. PE is caused by both maternal and fetal/placental factors. Placental vessels develop abnormally early in pregnancy and result in a placental hypoperfusion with the release of antiangiogenic factors into the maternal circulation. These factors promote endothelial dysfunction, which leads to the development of a vascular leak that enhances the volume expansion. MBG causes endothelial hyperpermeability, activating MAPKs with the disruption of the tight junction. This feature triggers apoptotic mechanisms, resulting in further endothelial dysfunction and leading to edema and the release of angiogenic factors [47,146].

An important element that could make MBG a potential biomarker for the early detection of PE was shown in an animal study, in which circulating MBG was shown to rise earlier than the onset of proteinuria and hypertension [13]. Thus, MBG can represent a predictive marker of PE, opening up new perspectives for its prevention, as well as halting its progression to the severe clinical state of eclampsia. Moreover, the administration of MBG in pregnant rats mimics this syndrome with proteinuria, hypertension and intra-uterine growth restriction (IUGR) [12,147].

As well as MBG being involved in the pathophysiology and progression of PE, its antagonist RBG was seen to have a role in the prevention and possible treatment of this disorder [148,149]. The administration of RBG in a rat model of PE leads to the resolution of hypertension and, if given early in pregnancy, it prevents all symptoms of PE, including IUGR [112]. Recently, a relationship between MBG and leptin in a pregnancy model of Sprague Dawley rats was investigated, showing that RBG administration can reverse the leptin-induced increase in systolic blood pressure, proteinuria and endothelial activation and suggesting a link between MBG and leptin signaling during pregnancy [150]. The RBG and, consequently, MBG molecular actions were also related to oxidative stress pathways [151], further amplifying the network of the interactions of CTS.

Similar to the action of RBG, Fedorova at al. investigated the role of antibodies against MBG in reversing the placenta-induced fibrosis of umbilical arteries in PE [152]. Monoclonal anti-MBG antibodies ex vivo reversed the placenta-induced fibrosis of umbilical arteries, indicating an active role of MBG in placental pathology. This represents the starting point on the possible development and use of RBG or monoclonal antibody Fab fragments to MBG [113] for the prevention and/or treatment of this complex syndrome.

Likewise, mineralocorticoid antagonists have been demonstrated to block the development of the fibrosis of umbilical arteries in PE, which is likely related to elevated MGB plasma levels [153].

Increased MBG production in women with PE, compared with physiological pregnancies, has also been demonstrated in humans [14,26]. Indeed, in healthy pregnant women, MBG levels are twice as high as in non-pregnant controls [26], with an increase in up to eight times in patients with PE [27], which leads to an increase in blood pressure values, due to direct vasoconstriction and profibrotic changes that occur at the umbilical and placental level. MBG also alters cytotrophoblast differentiation in the first trimester of gestation [16], suggesting its involvement in the early pathological events leading to PE. Human cytotrophoblast cells cultured in the first trimester and stimulated by MBG have been shown to undergo alterations of proliferation, migration and invasion, caused by the activation of JNKs, p38 and SRC and leading to increased cell apoptosis [15].

The complete and definite role of MBG in the pathogenesis of PE is not yet fully known. In addition, the mechanisms and sites of synthesis of this molecule in mammals are also not completely understood, although it is already known that the placenta is a site for its production.

Beyond sex and gender differences, it was seen that racial differences may contribute to different clinical parameters involving endogenous CTS, such as sodium sensitivity [154], suggesting that these molecules are affected not only by sex- and gender-related factors but also by genetically related ones; further research is needed to define the set of MBG expression-regulating factors. A study by Kantaria et al. showed that salt-sensitivity of blood pressure values, in which CTS are implicated, may vary within the same population [155], suggesting a relevant impact of inter-individual variability on CTS activity.

## 7. MBG Action on the Nervous System

In addition to CV and renal effects, the action of sodium and MBG also occurs at the level of the central nervous system, through channels in the glial cells of the supraoptic and paraventricular nuclei, where the detection of osmolarity and salt takes place through a new isoform of the sodium Nax channels [156]. MBG also acts on Amyloid Precursor Protein (APP), apolipoprotein E (APOE) and Connective Tissue Growth factors (CTGFs). Specifically, APP triggers a cascade of neurodegenerative events, such as synaptic dysfunctions, the genesis of neurofibrillary tangles and neuronal death. New findings show that, after MBG infusion, downregulation of APP mRNA expression in vivo in DSS rats is present [17,157].

## 8. MBG Action on Cells of the Immune System

Interestingly, MBG also has an action on some cells of the immune system. For example, in one study [18], the role of MBG in acute inflammation was evaluated in a model of zymosan-induced peritonitis in vivo and in peritoneal macrophages in in vitro culture. Mouse models were treated for three days with either MBG at 0.56 mg/kg, saline or dimethyl sulfoxide, and at one hour after the last treatment, they were treated with 2 mg/mL zymosan. Next, peritoneal exudate was collected, and total and differential leukocyte numbers were evaluated. Among these cells, Zymosan-stimulated peritoneal macrophages showed cytokine changes in IL-6, IL-1β, and TNF-α plasma concentrations compared with the control group. Those treated with MBG at the lowest concentration had reduced plasma levels of IL-1β (45%), IL-6 (17%), and TNF-α (20%) compared to the zymosan group. In the other two groups, cytokine plasma levels remained similar to those of the control group. This demonstrated that MBG pre-treatment reduced neutrophil migration, probably due to alterations in vascular permeability. In addition, MBG showed no cytotoxicity to cultured peritoneal macrophages. As already mentioned, plasma concentrations of MBG that are not able to inhibit the enzyme Na^+^/K^+^-ATPase, however, can trigger the production of messengers and the activation of intracellular signaling pathways.

## 9. Conclusions

Altered (circulating or urinary) levels of MBG seem to predict the development of diseases characterized by volume expansion, such as CKD, certain CV diseases and PE, demonstrating its possible preventive, diagnostic or monitoring role. MBG can now probably be observed in a different “light”, as a new important target for the prevention/treatment of kidney and CV development and progression.

## Figures and Tables

**Figure 1 ijms-24-11186-f001:**
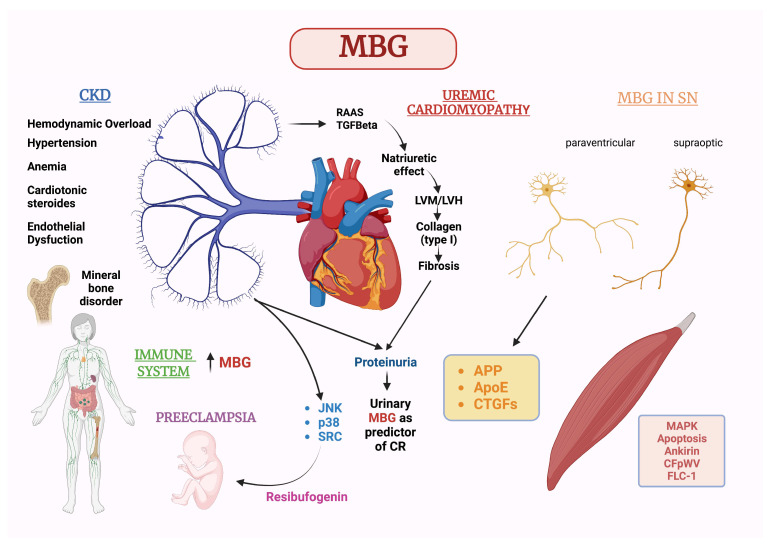
Main known pathways in the interactions between MBG and different organs and systems.

**Table 1 ijms-24-11186-t001:** Principal known effects of MBG on several organs and systems in Pre-clinical conditions.

Organs and Systems	Pathological Conditions	Effects of MBG	Pre-Clinical Studies	
Kidney and CV system	Volume-expanding conditions: essential hypertension, heart failure, PE, CKD complications: LVH, UC, myocardial fibrosis, diastolic dysfunction	Sodium and fluid retention, organ fibrosis and remodeling, vascular and microcirculation alterations, activation of oxidative stress pathways	**Model**	**Results**
			Fedorova, O.V., et al. [4]; male Fisher 344XNB rats and anesthetized dogs	IV saline infusion to anesthetized rats induced a significant increase in MLF plasma levels and pituitary OLC. No changes in pituitary MLF levels. Two hours of plasma volume expansion in anesthetized dogs increased urinary release of MLF. No change in OLC immunoreactivity. Evidence for the presence of a bufadienolide EDLF in mammals. Volume expansion stimulates EDLF response with the stimulation of brain OLC and plasma bufadienolide.
			Bagrov, A.Y., et al. [5]; anesthetized dogs	Decreased urinary release of MLF after volume expansion, but no changes in OLC, suggesting a bufadienolide nature of mammalian EDLF.
			Kennedy, D.J., et al. [6]; male CD1 mice	Plasma MBG increased after PNx. PNx caused cardiac hypertrophy and fibrosis, inducing UC.
			Priyadarshi, S., et al. [7]; male Sprague Dawley rats, isolated cardiac myocytes	In rats subjected to remnant kidney surgery, the administration of green tea extract at the induced attenuation of LVH, hypertension and preserved cardiac Na-K-ATP-ase activity. In isolated cardiac myocytes, both MBG and ouabain increased ROS production, whereas the addition of green tea prevented the increase in ROS production.
			Kennedy, D.J., et al. [8]; male Sprague Dawley rats	Rats with PNx had a significant increase in MBG plasma levels and urinary excretion rates and developed UC.
			Elkareh, J., et al. [9]; male Sprague Dawley rats, isolated cardiac fibroblasts	PNx increased MBG levels. Heart tissue samples from rats subjected to MBG-infusion and PNx showed an important increase in collagen-1 and α smooth muscle actin, whereas immunization against MBG attenuated these effects. Cardiotonic steroids, such as MBG, play a substantial role in the pathogenesis of cardiac fibrosis.
			Xie, Z., et al. [10]; neonatal ventricular myocytes cultures	Ouabain increases ROS production in cardiac myocytes, while preincubation with NAC and vitamin E reduced these effects. In cultured myocytes, the effects of ouabain on growth and growth-related genes can be dissociated from its effect on the resting intracellular Ca2+, the latter being responsible for the positive inotropic effect of this drug. It remains to be clarified whether the redox state of the myocyte or the intact heart may alter the effects of cardiac glycosides on cardiac hypertrophy without affecting the positive inotropic effect of these drugs.
			Pamnani, M.B., et al. [11]; male Wistar rats	Bufalin infusion increased mean the arterial BP, HR and renal excretion of Na+ and water, while ouabain infusion in an equimolar dose produced a significantly smaller increase in these effects and had no effect on HR. Bufalin has a greater effect on CV contractility and renal excretion of Na+ and water in rats than ouabain.
Feto-placental unit	Sex and gender medicine: pregnancy diseases, PE	Endothelial dysfunction, apoptosis, release of angiogenic factors, umbilical arteries fibrosis	Vu, H.V., et al. [12]; pregnant female rats	MBG levels increased in pregnant female rats treated with deoxycorticosterone acetate and 0.9% saline compared with normal pregnant rats. The administration of MBG in normal pregnant female rats caused a significant increase in blood pressure and vasoconstrictive activity of uterine vessels, while no changes were observed with the infusion of ouabain or digoxin at the same concentration. There is a relationship between MBG and a PE-like syndrome in rats.
			Vu, H.V., et al. [13]; pregnant female rats	RBG administration reversed MBG effects on BP. Antagonism of MBG could be a future therapeutic strategy for PE.
			Agunanne, E., et al. [14]; pregnant female Sprague Dawley rats	RBG administration in early pregnancy prevented PE syndrome in a rat model. RBG also prevented IUGR and had no teratogenic effects. Treatment of PE can focus on drugs that do not compromise the fetus.
			Uddin, M.N., et al. [15]; human extra-villous CTB cell line SGHPL-4 derived from first trimester chronic villous tissue	MBG induces a negative effect on CTB cell function, including apoptosis. MBG has a role in abnormal placentation and altered vascular function typical of PE. Targeting the MBG signaling pathway may be a future therapeutic strategy in PE treatment.
			La Marca, H.L., et al. [16]; human extra-villous CTB cell line SGHPL-4 derived from first trimester chorionic villous tissues	MBG has an anti-proliferative effect on CTB before CTB differentiation into an invasive pathway. MBG inhibits CTB cell migration and growth factor-induced invasion processes. MBG expression in the early phase of pregnancy has a role in abnormal placentation and altered vascular function.
Nervous system	Synaptic dysfunction, genesis of neurofibrillary tangles, neuronal death	Genesis of new isoforms of sodium Nax channels	Grigorova, Y.N., et al. [17]; young Dahl salt sensitive rats	HS diet prohypertensive and profibrotic effects can be at least partially attributed to MBG increase. MBG and HS diet had similar effects on CV system. MBG and HS diet upregulated the expression of fibrosis and Alzheimer’s disease genes in LV of the rat model. Hippocampal neuronal density was not affected by MBG or HS diet. Brain plasticity in young rats probably helped the animals to sustain the MBG-induced central arterial stiffness, which is one of the underlying mechanisms of cognitive impairment.
Immune system	Inhibition of neutrophil migration, inhibition of pro-inflammatory cytokines	Anti-inflammatory activity in a dose-dependent manner	Carvalho, D.C.M., et al. [18]; Swiss mice peritoneal fluid	MBG inhibited polymorphonuclear leukocyte migration to the peritoneal cavity. MBG reduced the expression of different pro-inflammatory cytokines. MBG had no cytotoxicity effects on macrophages in peritoneum.

Legend: BP: blood pressure; CTB: cytotrophoblast; EDLF: endogenous digitalis-like factor; HR: heart rate; HS: high-salt; IUGR: intrauterine growth restriction; LV: left ventricle; LVH: left ventricular hypertrophy; MBG: marinobufagenin; MLF: marinobufagenin like-factor; NAC: N-acetylcysteine OLC: ouabain-like compound; PE: pre-eclampsia; PNx: partial nephrectomy; RBG: resinobufagenin; ROS: reactive oxygen species; UC: uremic cardiomyopathy.

**Table 2 ijms-24-11186-t002:** Principal known effects of MBG on several organs and systems in clinical conditions.

Organs and Systems	Pathological Processes	Effects of MBG	Clinical Studies	
			Population	Results
*Kidney and CV system*	Volume-expanding conditions: essential hypertension, heart failure, PE CKD complications: LVH, UC, myocardial fibrosis, diastolic dysfunction	Sodium and fluid retention, organ fibrosis and remodeling, vascular and microcirculation alterations, activation of oxidative stress pathways	Keppel, M.H., [19] et al.; in patients with arterial hypertension; plasma MBG levels were measured in 40 patients, of whom 11 patients had primary aldosteronism (PA) and 29 patients had essential hypertension after exclusion of PA.	MBG concentrations increased, but not significantly, and showed a direct correlation trend with albuminuria and proteinuria.
			Bolignano, D., [20] et al.; cohort of 29 patients on HD vs. healthy controls.	MBG levels in HD patients were significantly higher than in healthy controls and significantly reduced in HD patients experiencing IDH during follow-up. Inverse correlations were found between the absolute number of IDH episodes per person and, respectively pre-dialysis MBG, 2 h MBG and HD-end MBG. MBG levels remained basically unchanged in HD patients with no documented IDH episodes during follow-up.
			Piecha, G., et al. [21]; 68 HD patients vs. 68 age-, gender- and blood pressure-matched subjects without CKD.	Mean plasma MBG immunoreactivity was significantly higher in HD patients compared with subjects with normal kidney function. In HD patients, plasma MBG was higher in men than in women, while this difference was not observed in subjects with normal kidney function.
			Bolignano, D., et al. [22]; 46 HD patients vs. healthy controls.	MBG levels were significantly higher in HD patients than in healthy controls. A statistically significant trend in MBG levels was found across different patterns of LV geometry, with the highest values in eccentric LVH. MBG levels were higher in presence of diastolic dysfunction.
			Jablonski, K.L., et al. [23]; middle-aged/older adults with moderately elevated systolic BP, but otherwise free of CV disease, diabetes, kidney disease and other clinical disorders.	Urinary MBG excretion decreased after 5 weeks of low sodium diet compared with 5 weeks of high sodium, while plasma MBG levels were not different between sodium conditions. Urinary MBG excretion was related to urinary sodium excretion and blood pressure measurements
			Strauss, M., et al. [24]; young, apparently healthy Black and White adults (60).	A persistent positive association between carotid-femoral pulse wave velocity and MBG excretion was found in women but not in men. High endogenous MBG levels may contribute to large artery stiffness in women through pressure-independent mechanisms
			Strauss, M., et al. [25]; young, apparently healthy Black and White adults (63)	LV mass, end diastolic volume and stroke volume were positively related to MBG excretion. The relationship between LV mass and MBG excretion was evident in women but not in men. Women may be more sensitive to MBG effects on early structural cardiac changes
*Feto-placental unit*	Sex and gender medicine: pregnancy diseases, PE	Endothelial dysfunction, apoptosis, release of angiogenic factors, umbilical arteries fibrosis	Lopatin, D.A., et al. [26]; 6 non-pregnant women, 6 normotensive age-matched pregnant controls and 11 patients with PE.	MBG levels significantly increased in PE pregnant women. MBG induced a contractile response of isolated rings of human mesenteric arteries in a concentration-dependent manner. MBG has a pathogenic role in PE.
			Agunanne, E., et al. [14]; 17 pre-eclamptic women and 46 normotensive pregnant women in various gestational periods.	Serum and urinary levels of MBG were significantly greater in pre-eclamptic women than normotensive pregnant women. MBG can be used for prediction and diagnosis of PE.
			Nikitina, E.R., et al. [27]; 16 pre-eclamptic pregnant women and 14 gestational age-matched normal pregnant women.	Serum and urinary levels of MBG increased in pre-eclamptic women compared with normal pregnant women. MBG, through a Fli-1-dependent mechanism stimulates collagen synthesis in umbilical arteries, leading to the impairment of vasorelaxation. MBG may represent a potential target for PE therapy.

Legend: CKD: chronic kidney disease; CTB: cytotrophoblast; ESKD: end-stage kidney disease; Fli-1: friend leukemia integration 1 transcription factor; HD: hemodialysis; IDH: intradialytic hypotension; LV: left ventricle; LVH: left ventricular hypertrophy; MBG: marinobufagenin; PE: pre-eclampsia; UC: uremic cardiomyopathy. Specification: numbers in parentheses indicate the reference number in the text.

## Data Availability

Not applicable.

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
