# Peer review of "New Insights on the Role of Marinobufagenin from Bench to Bedside in Cardiovascular and Kidney Diseases"

_ijms, 2023, doi:10.3390/ijms241311186_

Round 1

Reviewer 1 Report

In this manuscript,  N Carullo et al. describe roles of Marinobufagenin during cardiovascular and kidney diseases and some actions on the nervous and the immune systems. The study appears to have been carefully performed, whoever, I have the following comments:

Major comments:

1.      The title is not clear. The authors should mention systems concerned by the roles of  Marinobufagenin. The sentence of the line 26 is an example.

2.      The manuscript should be completed by some tables and/or figures. The authors should present for example a figure summarizing the effect of Marinobufagenin on Na+ K+ ATPase.

3.      The extraction techniques are not important to be mentioned in the manuscript.

4.      The English of the manuscript should be checked by a native English scientist.

Minor comments:

1.      Abstract: The abbreviations CKD  and ESRD should not be mentioned since they are not used in this section.

2.      Introduction: line 26, this sentence should appear in the abstract section.

3.      Some abbreviations are mentioned but not used (for example: resibufogenin (RBG), uremic cardiomyopathy (UC))

4.      Line 31: two spaces between “ubiquitous in” & “cell membranes”. The same remark for line 49  and 422.

5.      Lines 153-154: cardiovascular is defined as CV but it is previously mentioned in the first sentence of the introduction.

6.      Line 189: CTS (endogenous cardiotonic steroids) is previously defined. In addition, the abbreviation should be placed between ().

7.      Line 307: ROS is already defined in line 89.

8.      Line 382: The _ between “medicine” & “with” should be removed.

9.      Line 460: The authors should remove ‘ after MGB in the subtitle

10.   Line 478: The authors should remove ().

Author Response

  1. The title is not clear. The authors should mention systems concerned by the roles of  Marinobufagenin. The sentence of the line 26 is an example. R: We changed the title as: "New insights on the role of Marinobufagenin from bench to bedside in cardiovascular and kidney diseases"
  2. The manuscript should be completed by some tables and/or figures. The authors should present for example a figure summarizing the effect of Marinobufagenin on Na+ K+ ATPase.  R: We inserted a figure and two table resuming the text.
  3. The extraction techniques are not important to be mentioned in the manuscript.
  4. The English of the manuscript should be checked by a native English scientist. R: Text was revised by a native English scientist

Minor comments:

  1. Abstract: The abbreviations CKD  and ESRD should not be mentioned since they are not used in this section. R:Corrected
  2. Introduction: line 26, this sentence should appear in the abstract section.R:Corrected
  3. Some abbreviations are mentioned but not used (for example: resibufogenin (RBG), uremic cardiomyopathy (UC))R:Corrected
  4. Line 31: two spaces between “ubiquitous in” & “cell membranes”. The same remark for line 49  and 422.R:Corrected
  5. Lines 153-154: cardiovascular is defined as CV but it is previously mentioned in the first sentence of the introduction.R:Corrected
  6. Line 189: CTS (endogenous cardiotonic steroids) is previously defined. In addition, the abbreviation should be placed between ().R:Corrected
  7. Line 307: ROS is already defined in line 89. R:Corrected
  8. Line 382: The _ between “medicine” & “with” should be removed.R:Corrected
  9. Line 460: The authors should remove ‘ after MGB in the subtitleR:Corrected
  10. Line 478: The authors should remove ().R:Corrected

Reviewer 2 Report

Dear Authors,

Excellent comprehensive review paper explaining in detail as well as summarising about the different experiments and findings on MBG.

I suggest if you can include some pictorial representation of the MBG mechanisms of action at the renal levels and also how Sodium and renin angiotensin aldosterone pathway affect the MBG levels, it will be easier for the readers to understand in better about this. 

Also, I am interested to know if there is any research done on how the MBG levels based on the different ethnicities like caucasians, African Americans, Asians. Please include that as well to see if you find any data on this. 

Author Response

Dear reviewer,

thanks for your comments. We inserted a figure and two tables resuming the text. We haven't found studies on MBG levels based on the different ethnicities like caucasians, African Americans, Asians. It could be an idea for future investigations.

Kind regards

Round 2

Reviewer 1 Report

I would like to thank the authors for their corrections in the last revision of the manuscript. However, I have other remarks as indicated in the following points.

Line 17: The authors defined two abbreviations that are not used in the abstract. They should not have needed to use them.

Line 155: The authors referred to "cardiovascular" as "CV." However, this word appears in the text without any abbreviation (line 2, 38, 260, 279, …). The authors should define any abbreviation in the first use.

Line 153: The authors should replace 13,6% by 13.6%.

Line 187: "βeta" should be indicated as "β" or "Beta".

Lines 187, 317, and 318: The authors should indicate abbreviations in parentheses. For example, "phosphatidylinositol 3-kinase (PI-3K)" instead of "PI-3K (phosphatidylinositol 3-kinase)".

Line 304: The sentences from 304 to 311 are difficult to understand.

Line 313: "Ca++" should be indicated as "Ca2+".

Line 341: The authors should review the following sentence: "When FLI-1 is phosphorylated by phosphorylated Protein kinase C delta type (PKC-δ), it releases the procollagen DNA promoter." It is not clear to understand.

Author Response

Dear reviewer thanks for your comment that permitted to improve the manuscript!

Line 17: The authors defined two abbreviations that are not used in the abstract. They should not have needed to use them.

Corrected!

Line 155: The authors referred to "cardiovascular" as "CV." However, this word appears in the text without any abbreviation (line 2, 38, 260, 279, …). The authors should define any abbreviation in the first use.

Corrected!

Line 153: The authors should replace 13,6% by 13.6%.

Corrected!

Line 187: "βeta" should be indicated as "β" or "Beta".

Corrected!

Lines 187, 317, and 318: The authors should indicate abbreviations in parentheses. For example, "phosphatidylinositol 3-kinase (PI-3K)" instead of "PI-3K (phosphatidylinositol 3-kinase)".

Corrected!

Line 304: The sentences from 304 to 311 are difficult to understand.

Corrected!

Line 313: "Ca++" should be indicated as "Ca2+".

Corrected!

Line 341: The authors should review the following sentence: "When FLI-1 is phosphorylated by phosphorylated Protein kinase C delta type (PKC-δ), it releases the procollagen DNA promoter." It is not clear to understand.

Corrected!

Reviewer 2 Report

Thanks for making the changes as requested. 

Author Response

Thanks for your comments that permitted to improve our manuscript!